# A Small Molecule That In Vitro Neutralizes Infection of SARS-CoV-2 and Its Most Infectious Variants, Delta, and Omicron

**DOI:** 10.3390/biomedicines11030916

**Published:** 2023-03-15

**Authors:** Arfaxad Reyes-Alcaraz, Hanan Qasim, Elizabeth Merlinsky, Glenn Fox, Tasneem Islam, Bryan Medina, Robert J. Schwartz, John W. Craft, Bradley K. McConnell

**Affiliations:** 1Department of Pharmacological and Pharmaceutical Sciences, College of Pharmacy, University of Houston, Houston, TX 77204, USA; areyesa2@central.uh.edu (A.R.-A.);; 2Rogers State University, 1701 W. Will Rogers Blvd., Claremore, OK 74017, USA; 3Department of Biology and Biochemistry, University of Houston, Houston, TX 77204, USA

**Keywords:** SARS-CoV-2 spike protein, COVID-19, Delta, Omicron

## Abstract

The COVID-19 pandemic has underscored the urgent need to develop highly potent and safe medications that are complementary to the role of vaccines. Specifically, it has exhibited the need for orally bioavailable broad-spectrum antivirals that are able to be quickly deployed against newly emerging viral pathogens. The Severe Acute Respiratory Syndrome Coronavirus 2 (SARS-CoV2) and its variants Delta and Omicron are still a major threat to patients of all ages. In this brief report, we describe that the small molecule CD04872SC was able to neutralize SARS-CoV2 infection with a half-maximal effective concentration (EC50) = 248 μM. Serendipitously, we also were able to observe that CD04872SC inhibited the infection of the SARS-CoV-2 variants; Delta (EC50 = 152 μM) and Omicron (EC50 = 308 μM). These properties may define CD04872SC as a potential broad-spectrum candidate lead for the development of treatments for COVID-19.

## 1. Introduction

The outbreak of coronavirus disease 2019 (COVID-19) caused by the severe acute respiratory syndrome coronavirus 2 (SARS-CoV-2/2019-nCoV) poses a serious threat to global public health and local economies. As of March 11, 2023, over 759 million cases have been confirmed around the world [1]. Such huge numbers of infected individuals and their associated mortality call for an urgent demand for effective, available, and affordable drugs to control and diminish the pandemic. Pfizer’s novel COVID antiviral treatment is only useful during the first three days of showing symptoms [2]. Pfizer’s PAXLOVID antiviral medication (Appendix A) is not expected to be widely available for treating a large number of patients in a timely manner. The last SARS-CoV-2 variant (Omicron) is rapidly spreading and is more contagious than the original SARS-CoV-2 strain, as well as the SARS-CoV-2 Delta variant [3]. The SARS-CoV-2 variants (Delta and Omicron) demonstrate how easily this virus can accommodate antigenic changes in its spike (S) protein without the loss of fitness [4]. The Omicron variant has particularly stressed healthcare systems around the world. Therefore, identifying effective antiviral agents to combat this infectious disease is urgently needed.

## 2. Methods

### 2.1. Molecular Modeling

Following well-established protocols, structures from the protein data bank (PDB files 6M0J, 6VXX, and 6VYB) were used to prepare the structural models of the SARS-CoV-2 spike protein receptor binding site and the ACE2 receptor [5,6]. CD04872SC was initially expanded into its 3D minimized structure using BALLOON 3-1.6.6 and then processed with PRODRG4 to build topology files [7,8]. Both the spike protein and the ligand were prepared for docking analysis using AUTODOCK-TOOLS version 4 and AUTODOCK_VINA 1.1.2. Docking for each of the millions of compounds evaluated from the Maybridge and ZINC library was completed with the exhaustiveness set to 100, and the top 20 poses evaluated in each tranche and evaluated in both the Microsoft Azure Cloud and on our university servers.

### 2.2. Lentivirus-Mediated Expression of the Spike Protein of SARS-CoV-2

All manipulations always took place in a biosafety cabinet. HEK293 cells were transfected with the plasmids containing the SARS-CoV-2, Wuhan-Hu-1 (GenBank: NC_045512) spike-pseudotyped lentiviral kit (NR-52948, from Bei Resources (Manassas, VA, USA) designed to generate pseudotyped lentiviral particles expressing the spike (S) glycoprotein gene, as well as luciferase (Luc2) and ZsGreen. Seventy-two hours after transfection, the medium was collected in a 50 mL tube and stored at −80 °C for further applications. This protocol only requires Biosafety Level 1 (BSL1) conditions, and the viruses used in this protocol were replication-defective.

The plasmid expressing the human ACE2 gene was acquired from Addgene (Cat. No. 1786). The plasmid expressing the Delta Spike Protein (Cat. No. VG40819-UT) was acquired from Sino Biological (Beijing, China). The plasmid expressing the Omicron Spike Protein (Cat. No. MC_0101274) was acquired from GenScript (Piscataway, NJ, USA).

The following describes the detailed protocol for the generation of pseudo typed lentiviral particles by transfecting HEK293. We used the following protocol from Crawford and colleagues [9]. In brief, we seeded HEK293 cells in a DMEM high glucose growth medium so that they would be 70% confluent the next day. For a six-well plate, this was equivalent to 8 × 10^5^ cells per well. The next morning, following 12–16 h of seeding, the cells were transfected with the plasmids required for lentiviral production. We transfected the cells using Viafect (Promega Corporation, Madison, WI, USA, Cat. No. E4982) following the manufacturer’s instructions and using the following plasmid mix per well of a six-well plate (plasmid amounts were adjusted for larger plates):

1 µg of lentiviral backbone: ZsGreen (NR-52520) or the luciferase-IRES-ZsGreen (NR-52516) backbone;

0.22 µg each of plasmids: HDM-Hgpm2 (NR-52517), pRC-CMV-Rev1b (NR-52519), and HDM-tat1b (NR-52518);

0.34 µg viral entry protein: either SARS-CoV-2 Spike (NR-52513, NR-52514);

At 60 h post-transfection, the virus was collected by harvesting the supernatant from each well and filtering it through a 0.45 µm SFCA low protein-binding filter. The viruses were then concentrated by using Lenti-X ^TM^ (Cat. No. 631231, Takara Bio Inc., San Jose, CA, USA). The virus was resuspended in 200 µL of PBS and followed by the viral suspensions stored frozen at −80 °C.

### 2.3. Neutralization Assay

We seeded a poly-L-lysine-coated 96-well plate with 4 × 10^4^ HEK293 cells per well in 100 µL DMEM High Glucose. The following morning, we transfected the cells with Viafect (Promega, Cat. No. E4982) and 100 ng of ACE2 plasmid per well. We let the ACE2 receptor express in the cells for 36 h after transfection. We then pretreated the cells with different concentrations of the different compounds and incubated the cells for 1 h. We then used 10 µL concentrated viral suspension to infect the cells in 96 well plates. We incubated them at 37 °C for 36 h before measuring the green fluorescence (530/30 filter to detect ZsGreen in the FITC channel). For our analysis, we first subtracted out the background signal (average of the “virus only” and “virus + HEK293” wells) and then calculated the “maximum infectivity” for each plate as the average signal from the wells without the drug (“virus + cells” wells). We then calculated the “fraction infectivity” for each well, as the green fluorescence reading from each well was divided by the “maximum infectivity” for that plate. For the curves shown in Figure 1, we then fit and plotted the fraction infectivity data using Prism 7. This program fits a three-parameter Hill curve to determine the potency of neutralization (Equation (1); below), with the top baseline being a free parameter and the bottom baseline fixed to zero.

### 2.4. Thermal Shift Assay

To monitor the SARS-CoV-2 spike protein unfolding, the Protein Thermal Shift Dye was used. Protein Thermal Shift Dye is an environmentally sensitive dye. The unfolding process exposes the hydrophobic region of proteins and results in a large increase in fluorescence, which is used to monitor the protein-unfolding transition. The thermal shift assay was conducted in the CFX Opus 384 Real-Time PCR System (Bio-Rad, Hercules, CA, USA) originally designed for PCR. The system contains a heating/cooling device for accurate temperature control and a charge-coupled device (CCD) detector for the simultaneous imaging of the fluorescence changes in the wells of the microplate. Solutions of 20 μL were prepared as follows: 5 μL of Thermal Shift Assay buffer + 12.5 μL of viral particles in the absence or the presence of CD04872SC + 2.5 μL Diluted Thermal Shift Dye (8X) were added to each well of the 384-well PCR plate. The final concentration of CD04872SC was 500 μM. The plate was heated from 20 to 95 °C with a heating rate of 0.5 °C/min. The fluorescence intensity was measured using the ROX channel.

### 2.5. Statistics and Reproducibility

The results were analyzed using the Prism 7 application (Graph Pad Software Inc., San Diego, CA, USA). Dose-response curves were fitted using the following three-parameter equation:(1)$$Response=Bottom+\frac{Top-Bottom}{1+10^{\left(logEC_{50}-\log\left[A\right]\right)}}$$

In this equation, the *Bottom* and *Top* are the lower and upper plateaus, respectively, of the concentration-response curve and where [*A*] is the molar concentration of the agonist and EC50 is the molar concentration of the agonist required to generate a response halfway between the top and the bottom.

## 3. Results

### 3.1. In Silico Screening

Since the SARS-CoV-2 virus enters host cells via an interaction between its spike protein and the host cell receptor angiotensin-converting enzyme 2 (ACE2) [10], our goal was therefore to disrupt this interaction with a small molecule. To disrupt this protein-protein interaction, we performed in silico screening following our Cloud workflow implementation using Microsoft Azure and the University of Houston’s Hewlett Packard Enterprise (HPE) Data Science Institute to execute our drug discovery workflow [7,8] by using AutoDock Vina version 1.2.0, GROMACS 2021.5, and PySpark 3.3.2. Candidate in silico binding poses of compounds to the SARS-CoV-2 spike protein were generated from over a million compounds from the Maybridge and ZINC libraries. We selected the top 15 molecules through in silico screening, that disrupted the interaction between the spike protein and the ACE2 receptor. Molecular dynamic simulations revealed that some of the compounds from these libraries had favorable interactions with the spike protein’s ACE receptor binding domain interface, leading to a potential neutralization of the SARS-CoV-2 infection. One of those compounds (CD04872SC) formed a close association between its amide carbonyl of the bicyclic indolin-2-one group and the backbone N of GLY169 (3.1 Å), and has hydrophobic interactions with TYR116, TYR162, and TYR172 (Appendix A). The Lipinski and Muegge rules; MLOGP > 4.15 (4.22 actual prediction) and XLOGP3 > 5 (5.73 actual prediction), respectively, were used as guidelines for the drug-likeness of initial leads. Our compound slightly exceeded these guidelines, but it remained a viable candidate.

### 3.2. Inhibition of In Vitro Infection of SARS-CoV2, Delta, and Omicron

To corroborate our in-silico data, cell infection and viability assays were carried out to measure the effects of these top 15 compounds on cell infection rates and cell cytotoxicity. HEK293 cells (ATCC-1586) overexpressing the ACE2 receptor were infected with lentiviruses that express the spike protein found in SARS-CoV-2 [9] (Figure 1a). Next, in the presence of the compounds identified from the in silico data, we measured the green fluorescence intensity of ZsGreen1 protein 48 h post-infection. Next, this assay was repeated by independently expressing the Spike protein’s variants, Delta and Omicron, also in HEK293 cells overexpressing the ACE2 receptor (Figure 1a). These in vitro SARS-CoV-2 spike proteins and ACE2 receptor co-expression cellular studies confirmed the in-silico molecular strategy that aimed at disrupting the interface of the SARS-CoV-2 spike protein and ACE2 receptor by docking a small molecule at the receptor-binding site of the spike protein (Figure 1b). We then performed infection inhibition drug screening and cell cytotoxicity assays. From the drug screening assays, we identified CD04872SC as our lead compound (Figure 1c). We then validated this lead compound by determining its infection rate. We found that CD04872SC showed a half-maximal effective concentration (EC50) = 248 μM; the concentration capable of reducing the viral infection of the cells expressing the SARS-CoV-2 spike protein (Figure 1d). Furthermore, we were able to observe that CD04872SC also inhibited the infection of the SARS-CoV-2 variants; Delta (EC50 = 152 μM) and Omicron (EC50 = 308 μM) (Figure 1d). Finally, this CD04872SC lead compound was then tested at various concentrations in cell cytotoxicity assays using the PrestoBlue™ Cell Viability Reagent (Life Technologies, Carlsbad, CA, USA, Cat. No. A13261) kit, and we found that CD04872SC was without major cell cytotoxicity within the indicated concentrations (Figure 1e). It would be interesting to study the potential cytotoxicity in other cell models such as the endothelial, hepatic, and renal cells.

### 3.3. CD04872SC Inhibits Viral Infection by Direct Binding to the Spike Protein of SARS-CoV2, Delta, and Omicron

A Protein Thermal Shift assay was developed for the analysis of viral particles’ melting point fluorescent readings directly from a BIO-RAD real-time PCR instrument. Viral particle stability changed in the presence of CD04872SC. Real-time melt experiments were used to demonstrate the direct binding between CD04872SC and the spike protein of each SARS-CoV2 variant. The presence of CD04872SC binding to the spike protein is evaluated as changes in the fluorescence profiles (melt curves), as shown in Figure 2. This was converted to a T_m_, which is calculated based on the inflection point of the melt curves.

The potential mechanisms of action of CD04872SC for the prevention and treatment of SARS-CoV-2 infection are described in Figure 2a. In the absence of specific drugs, SARS-CoV-2 binds the ACE2 receptor through the receptor binding domain (RBD) in the S1 subunit, mediating viral entry and subsequent membrane fusion. In the presence of CD04872SC targeting and binding to the SARS-CoV2 RBD, interaction with the ACE2 receptor is blocked and membrane fusion is inhibited. To demonstrate the binding between CD04872SC and the spike proteins of each variant, we performed a thermal shift assay (Figure 2). Thermal shift assays measure changes in the thermal denaturation temperature, serving as an indicator of the stability of a protein under varying conditions such as when bound by a drug, buffer pH, ionic strength, redox potential, or sequence mutation. The method for measuring spike protein thermal shifts was based on differential scanning fluorimetry (DSF) or thermofluor, which utilizes a specialized fluorogenic dye. The binding of low molecular weight ligands can increase or decrease the thermal stability of a protein, as described by Daniel Koshland (1958) [11]. As shown in Figure 2, we observed a difference of about 3 °C of stability in the presence of CD04872SC in the SARS-CoV2 viral suspensions versus its absence. Moreover, we were able to observe the same stabilizing tendency for the variants Delta and Omicron (Figure 2).

## 4. Discussion

Our findings reveal that CD04872SC (with a molecular weight of 423.3 Da and theoretical logP of 3.11 [12], Appendix A) is effective in inhibiting the SARS-CoV-2 spike protein binding to the ACE2 receptor, including its most infectious variants (Delta and Omicron) in functional in vitro assays. To the best of our knowledge, this small molecule has not been involved in any clinical trials as of yet. These data were developed from the application of the large-screen computational screening of small molecule databases for hits against the spike protein target and ranked for evaluation in cell-based assays. A SARS-CoV-2 specific reporter system was used by our group and used to confirm that our small molecule lead, CD0487SC, did prevent the binding with the ACE receptor and will form the foundations of a drug development effort to find derivatives and enhancements to evolve CD0487SC into a clinical viable drug candidate. However, our drug candidate needs further validation in another biological context since it has recently been reported that other several membrane proteins in addition to ACE2 have been proposed to be alternative receptors for SARS-CoV and SARS-CoV-2. For instance, CD147 has been suggested to serve as an alternative receptor for SARS-CoV and SARS-CoV-2 infection [13,14]. In addition, NRP1 was shown to enhance the TMPRSS2-mediated entry of wild-type SARS-CoV-2 but not that of the mutant virus that lacks the multibasic furin-cleavage site [15]. NRP1 also can bind to S1 through the multibasic furin-cleavage site to promote S1 shedding and to expose the S2′ site to TMPRSS2 [16].

In contrast to vaccines, which usually take several weeks to induce antibody production in immunized individuals [17], neutralizing small molecules may provide immediate protection against viral infection, and thus be suitable for people of all ages, and may be particularly suitable for high-risk populations and immunocompromised individuals who typically do not generate sufficient antibodies after vaccination. Other available strategies include harvesting patient plasma from convalescent patients with potent neutralizing activities. Although the United States Food and Drug Administration has issued an emergency use authorization for the application of convalescent plasma to treat hospitalized patients with COVID-19, more randomized clinical trials are needed to determine the real efficacy and safety of convalescent plasma [18]. Another approach is by producing effective neutralizing monoclonal antibodies (mAbs), but doing so in a cost-effective manner, and at scale, is very challenging; most mAbs are produced in mammalian cells with relatively low productivity and thus high production costs. Manufacturing approaches that produce effective neutralizing antibodies with high expression yield at low cost are acutely needed.

Many treatment approaches are in development, with outcomes remaining to be accessed. Now is the time to press the development of numerous leads and mature the technologies of each candidate lead. Here, we have shown that the small molecule CD0487SC is able to bind and neutralize the SARS-CoV-2 Spike protein, preventing infection. The application of preclinically tested small molecules to patients with COVID-19 requires intensive study and testing. Currently, the hope is that antibody cocktails specific to different neutralizing epitopes in the SARS-CoV-2 RBD, or other regions in the S protein of SARS-CoV-2, make effective COVID-19 treatments and provide short-term protection against viral infection. However, as leads like CD0487SC are developed, more options for treatment will improve the likelihood of sustainable interventions against the SARS-CoV-2 pandemic. In addition, the possibility that this drug candidate may also interact with other S proteins from other coronaviruses remains. This may result in offering wider protection against other viral infections by various types of coronaviruses.

In summary, this study suggests that CD04872SC might be the starting point for the potential treatment of severe COVID-19 and mortality in the era of Omicron. However, further candidate lead development of derivatives and preclinical studies such as testing CD04872SC and its derivatives’ effectiveness in animal models are still needed. Currently, the COVID-19 vaccine remains the most effective intervention to prevent disease progression and death among COVID-19 patients. However, this promising drug candidate lead should be developed into a family of derivatives that could be further refined, possibly leading to a more efficacious and cost-effective alternative to expensive neutralizing treatments based on monoclonal antibodies.

## Figures and Tables

**Figure 1 biomedicines-11-00916-f001:**
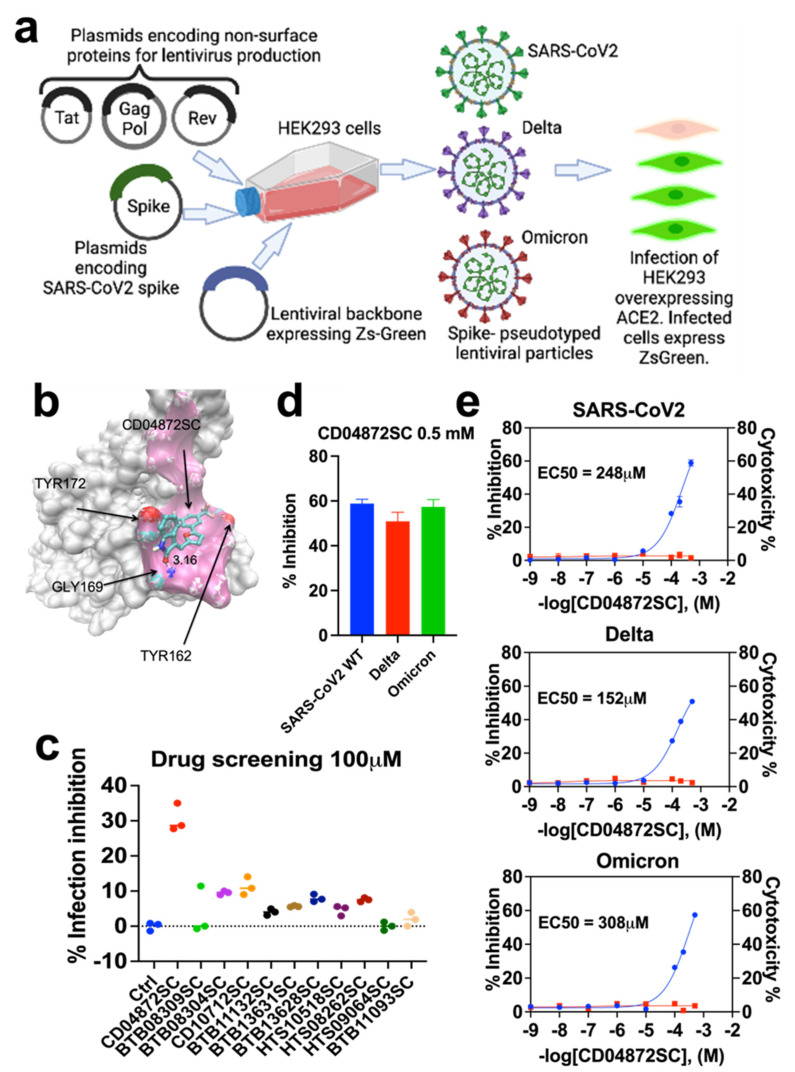
Lentiviral system approach and antiviral activities of the test drugs against SARS-CoV-2 and its variants Delta and Omicron in vitro. (**a**) HEK293 cells were transfected with a plasmid encoding a lentiviral backbone (genome) expressing a marker protein, a plasmid expressing spike proteins of SARS-CoV-2 and its variants Delta and Omicron, and plasmids expressing the other proteins needed for virion formation (Tat, Gag-Pol, and Rev). The transfected cells produced lentiviral particles with SARS-CoV-2 spikes on their surface. These viral particles can infect cells that overexpress the ACE2 receptor. The readout to measure the grade of viral infection is the green fluorescence of ZsGreen1 from infected cells. (**b**) The molecular structure shows the interface of the receptor-binding domain (RBD) of the SARS-CoV-2 spike protein (white surface) with the human ACE2 receptor (pink surface). CD04872SC is predicted to bind on the face leading to the potential disruption of the complex. (**c**) Drug screening of the selected top compounds (each at a concentration of 100 µM) was evaluated for their antiviral activity against SARS-CoV-2, as defined by their percent infection inhibition. (**d**) The maximum antiviral activity against SARS-CoV-2, and its two major variants Delta and Omicron, for lead drug candidate CD04872SC. (**e**) The mean percent inhibition of viral infection and percent cell cytotoxicity of lead drug candidate CD04872SC in HEK293 cells expressing the ACE2 receptor. In brief, HEK293 cells overexpressing the ACE2 receptor were pre-treated with the lead drug candidate CD04872SC for 1 h. Next, 10 µL of the lentiviral suspensions expressing the spike protein was then added to the culture plates and incubated for 48 h. After the incubation period, the green fluorescence of ZsGreen1 protein was measured with the plate reader Synergy 2 (BioTek, Winooski, VT, USA) using the FITC channel. The mean percent inhibition and percent cytotoxicity is represented on each of the graph’s left and right y-axis, respectively. For each of the three graphs, blue curves indicate the percent of infection inhibition and red curves represent the percent of cytotoxicity. The results are expressed as mean ± s.e.m. of three experiments performed in triplicate; each triplicate was averaged before calculating the s.e.m.

**Figure 2 biomedicines-11-00916-f002:**
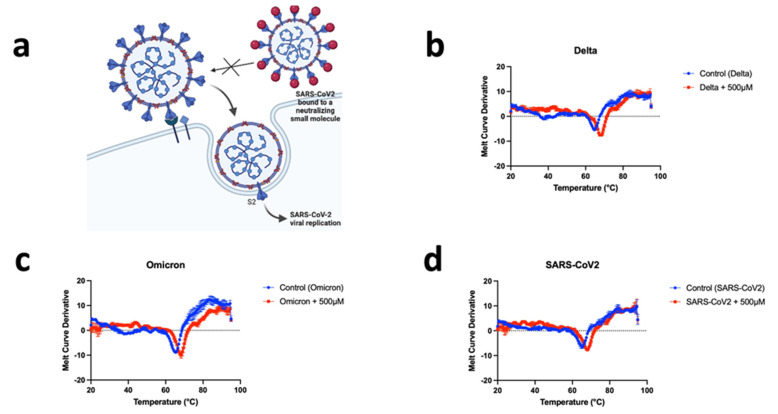
CD04872SC binds to the spike protein of SARS-CoV-2, Delta, and Omicron. (**a**) Schematic representation of the viral neutralization concept of SARS-CoV2 by a small molecule. Targeting a specific region of the spike proteins makes it possible to neutralize the coronavirus by disrupting the binding between its spike protein and receptor ACE2. (**b**) The thermal stability of the Delta variant pseudotyped lentiviral particles in the absence and presence of 500 μM CD04872SC. (**c**) Thermal shift assay performed with Omicron variant pseudotyped viral particles in the absence and presence of 500 μM CD04872SC. (**d**) Thermal stabilization of SARS-CoV2 pseudotyped viral particles in the presence of 500 μM CD04872SC. The results are expressed as the mean of three experiments performed in triplicate; each triplicate was averaged before calculating the s.e.m. Lead compound CD04872SC is part of a “U.S. Provisional Patent Application No. 63/312,351, filed on Monday, 21 February 2022”.

## Data Availability

Underlying data is available using the following link: https://figshare.com/search?q=10.6084%2Fm9.figshare.19522198 (accessed on 4 May 2022) and from the authors upon request.

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
