# Peer review of "A Small Molecule That In Vitro Neutralizes Infection of SARS-CoV-2 and Its Most Infectious Variants, Delta, and Omicron"

_biomedicines, 2023, doi:10.3390/biomedicines11030916_

Round 1
Reviewer 1 Report
Overall, it is well written manuscript with all relevant experimental details provided. I have no reservations except that I recommend that the authors provide the chemical structure of the identified molecule, and provide the chemical structures of other entry inhibitors that have been in use or under clinical testing for COVID-19.
Author Response
Thanks so much for your valuable comments to our manuscript. Following your suggestion, we have included more details of the lead molecule as well as other COVID-19 treatments currently in the market in Supplementary Information. We also have included the corresponding molecular weight (Please see discussion section, lines 259).
Reviewer 2 Report
The introduction does not fully reflect the problem, in particular, the authors reduce the interaction of the virus exclusively with ACE-2, although it is known that this is far from the only “receptor”, and others should be mentioned like CD147, CD209 etc. In addition, the spectrum of cells expressing both ACE-2 and other coronavirus “receptors” is not reflected. More serious problems follow from this. So, the authors do not give not only the structure of the CD04872SC substance (what yet can be understood: a commercial secret, etc.), but even elementary physicochemical characteristics, such as: molecular weight, LogP and redox potential, by which one could roughly estimate bioavailability and toxicity (although there are special computer programs for this, probably known to the authors, the use of which would not lead to the disclosure of the secrets). But even information about effective concentrations allows the specialist to conclude that the potential dosage in in vivo experiments is extremely high, which will be associated with high toxicity to the body. 250 μM at a molecular weight of about 400 gives a figure of 100 mg/kg at 100% bioavailability, and if we make a correction for the actual bioavailability of drugs, then this figure should be increased by 5-10 times. As a result, we will receive a dose of the drug, which will most likely have a toxic effect on the liver and on the whole organism. The absence of a toxic effect for the transformed HEK293 cell line does not mean the absence of such an effect for much more sensitive hepatocytes or endothelial cells. By the way, tests for cytotoxicity associated with the determination of dehydrogenase activity have a serious drawback in the case of testing compounds with a relatively high or low redox potential, because upon activation of transmembrane dehydrogenases or direct interaction with the probe, false positive or false negative effects can be obtained. The authors mention van der Waals interactions in the manuscript, but in the absence of objective data, this statement looks unfounded. Therefore, when solving such an important problem as cytotoxicity, it is necessary to use other or additional test systems, for example, with neutral red or the determination of ATP in cells. Moreover, even in the absence of a toxic effect for e.g. hepatocytes, this does not mean the absence of absorption of the substance by cells and their biotransformation, which entails a decrease in the bioavailability of the drug.
There are semantic typos in the text like “cells expressing SARS-CoV-2 Spike protein”.
Author Response
We really appreciate your valuable comments. We have made the following modifications to our manuscript addressing your concerns:
- We have included a paragraph mentioning other entry membrane receptors for SARS-CoV2, please see Discussion Section lines 266-272.
- We provide additional details of the lead compound in the Discussion Section lines 259 as well as in the Supplementary Information file.
- We have deleted the statement about Van der Waals interactions, please see line 121.
- We also corrected some sentences please see line 126.
- We have suggested that more work needs to be done regarding the cytotoxicity, such as to test the same lead compound concentrations in Vero cells, please see lines 212 and 213.
- We have also added an additional file to this manuscript (Supplementary Information) where you can find more information about the material and methods, other properties about our lead compound and other chemical 2D structures about COVID-19 treatments currently in the market.
Reviewer 3 Report
The manuscript shows an interesting approach to analyzing a small molecule, whose details are not provided, interaction with the S protein of the SARS CoV-2 virus interacting with the ACE2 receptor. Even though the manuscript, as stated by the authors, provides preliminary data, several questions were not answered: 1) the structure and function of the molecule, there are no supplemental files and figures as stated in the text, and 2) is this molecule used in any trial? Can it be safe to apply in mucosal tissue? The IC50 is very high, so the molecule must be modified chemically. What are the suggestions? It would be interesting to repeat the experiments in Vero cell cultures, The thermal assays do not provide strong evidence of the interaction between the compound and the S protein, and it is not specific. The molecule must probably bind to other S proteins from another virus of the coronavirus family. Finally the discussion should be modified accordingly.
Author Response
Thanks a lot for your very helpful observations. Following your suggestions, we modified our manuscript as follows:
- We have included a separate file as Supplementary Information where we provide more information about the material and methods as well as properties of the lead compound.
- To our best knowledge, this small molecule has not been involved in any clinical trial and we expressed this in lines 261 and 262 withing the manuscript.
- In lines 259 and 260 we provide the molecular weight as well as the calculated logP for this small molecule. One of the strategies to improve the potency is to increase the solubility by decreasing the logP value to the negative range. One way to do this would be by adding more fluor atoms.
- Yes, it might be that this compound interacts with Spike proteins from other types of coronaviruses. However, we have seen this as an advantage since it might mean potential protection against other viral infections by other coronaviruses. However, more work needs to be done to confirm this.
- The discussion has been changed accordingly to address all your concerns.
Round 2
Reviewer 2 Report
I cannot say that the authors have fulfilled all the recommendations, but the part that has been fulfilled allows us to admit possibility of the publication of this article and hope that the further direction of research will be corrected. Still, I would like to see the intentions of the authors to carry out experiments not only and not so much with the renal Vero line, but also with one of the hepatic lines, and especially with endothelial cells. I would also like to receive answers to my previous comments regarding high effective concentrations of CD04872SC, and indicate the theoretical redox potential in the list of physico-chemical characteristics.
Author Response
Thanks again for your valuable observations. We have added a statement indicating that in future experiments, we will test for potential cytotoxicity in other cell models, including endothelial, hepatic, and renal cell lines. Additionally, we have added a list of physicochemical and pharmacological properties list (Supplementary Table I) in the Supplementary information file where can find a wide range of theoretical properties of our drug candidate.
Reviewer 3 Report
The manuscript has been improved, and most of the queries were answered. The discussion was not modified as requested; however, it could be suitable for publication. Minor details in the text format should be corrected.
Author Response
Thank you so much for your great comments. The manuscript has been edited for have edited for grammar, readability, and format correctness.